Analysis of the fusion of multimodal sentiment perception and physiological signals in Chinese-English cross-cultural communication: Transformer approach incorporating self-attention enhancement

Bi Xin 1 bixin198409@163.com
Zhang Tian 2
1 School of Literature, Heilongjiang University , Harbin, Heilongjiang , China
2 Department of Languages and Literary Studies, Lafayette College , Easton, PA , United States
Asif Muhammad
Electronic publication date: 2025 May 23
Publication date: 2025
Volume: 11
Electronic Location ID: e2890
Received 2025 Jan 30; Accepted 2025 Apr 22
Copyright: © 2025 Bi and Zhang
Copyright year: 2025
Copyright holder: Bi and Zhang
License: This is an open access article distributed under the terms of the Creative Commons Attribution License, which permits unrestricted use, distribution, reproduction and adaptation in any medium and for any purpose provided that it is properly attributed. For attribution, the original author(s), title, publication source (PeerJ Computer Science) and either DOI or URL of the article must be cited.
License URL: https://creativecommons.org/licenses/by/4.0/

Keywords: MFCC, Cross cultural communication, Transformer, Information fusion

Funding: 2022 Heilongjiang Research Project on Teaching Reform of Undergraduate Education in Higher Education Teaching Reform and Practice of Teaching Chinese to Speakers of Other Languages Programme based on the Construction of Intercultural Competence: SJGZ20220055 This work is funded by the “2022 Heilongjiang Research Project on Teaching Reform of Undergraduate Education in Higher Education: Teaching Reform and Practice of Teaching Chinese to Speakers of Other Languages Programme based on the Construction of Intercultural Competence”, the project number is SJGZ20220055. The funders had no role in study design, data collection and analysis, decision to publish, or preparation of the manuscript.

==============================
With the acceleration of globalization, cross-cultural communication has become a crucial issue in various fields. Emotion, as an essential component of communication, plays a key role in improving understanding and interaction efficiency across different cultures. However, accurately recognizing emotions across cultural backgrounds remains a major challenge in affective computing, particularly due to limitations in multimodal feature fusion and temporal dependency modeling in traditional approaches. To address this, we propose the TAF-ATRM framework, which integrates Transformer and multi-head attention mechanisms for cross-cultural emotion recognition. Specifically, the framework employs bidirectional encoder representations from transformers (BERT) for semantic feature extraction from text, Mel-frequency Cepstral Coefficients (MFCC) and Residual Neural Network (ResNet) for capturing critical features from speech and facial expressions, respectively, thereby enhancing multimodal emotion recognition capability. To improve the fusion of multimodal data, the Transformer is utilized for temporal feature modeling, while multi-head attention reinforces feature representation by capturing complex inter-modal dependencies. The framework is evaluated on the MOSI and MOSEI datasets, where experimental results demonstrate that TAF-ATRM outperforms traditional methods in emotion classification accuracy and robustness, particularly in cross-cultural emotion recognition tasks. This study provides a strong technical foundation for future advancements in multimodal emotion analysis and cross-cultural affective computing.

Introduction

In today’s era of globalization, intercultural communication has become a vital means of fostering mutual understanding and cooperation between different countries and regions. As international cooperation and transnational mobility continue to grow, individuals from diverse cultural backgrounds engage in various interactions, spanning fields such as business, education, tourism, and socialization. However, differences in language, habits, and customs often present significant challenges in cross-cultural communication. Even when a common language is used, communicators from different cultures may misunderstand one another due to varying emotional expression styles (Shadiev, Wang & Huang, 2021). Emotions, therefore, play a pivotal role in cross-cultural communication. Accurate emotion recognition not only helps bridge language differences, but also enables a better understanding of the emotional attitudes of both parties in communication, avoiding misunderstandings or conflicts caused by emotional communication errors. From a psychological perspective, Ekman’s theory of basic emotions states that human basic emotions are universal across different cultures, but their expressions may vary. Meanwhile, Hofstede’s cultural dimension theory suggests that high context cultures (such as Japan and South Korea) rely more on nonverbal cues, while low context cultures (such as the United States and Germany) tend to express themselves directly. Therefore, cross-cultural emotion recognition not only needs to consider language differences, but also must combine multimodal information such as speech, facial expressions, and body language to accurately understand emotional expressions in different cultural backgrounds, thereby promoting more effective cross-cultural communication (Lan et al., 2021).

Multimodal sentiment recognition has emerged as a key research focus in affective computing, with the primary objective of enhancing sentiment recognition accuracy by analyzing data from multiple modalities (e.g., text, speech, vision, etc.). Traditional unimodal sentiment recognition methods, while effective in certain contexts, typically rely on a single modality. However, these unimodal approaches often fall short in capturing the full complexity of emotional expression, particularly when there are complementarities or contradictions among sentiment cues from different modalities (Abdullah et al., 2021). To address these limitations, multimodal emotion recognition methods have been developed, integrating information from multiple modalities to provide a more comprehensive understanding of emotional expressions. Significant progress in this area has been driven by advancements in deep learning techniques. For instance, convolutional neural networks (CNNs) are frequently employed to extract facial expression features from visual data, while long short-term memory (LSTM) are widely used for sentiment analysis of speech and text due to their ability to model time-series data effectively. Moreover, models based on the attention mechanism, particularly the Transformer, have shown excellent performance in multimodal sentiment recognition (Xie, Sidulova & Park, 2021). The Transformer model, with its self-attention mechanism, making it highly effective in multimodal sentiment analysis. In recent years, the multi-head attention mechanism has further enhanced the model’s ability to examine relationships between features from different modalities, especially when dealing with complex interactions in speech, text, and image data (Siriwardhana et al., 2020). These innovations in deep learning have provided substantial support for multimodal emotion recognition and have propelled advancements in the broader field of emotion computing research.

In light of the aforementioned research landscape and technological background, this article introduces an emotion recognition framework predicated on multimodal data fusion, termed TAF-ATRM. The central aim of this framework is to augment emotion recognition efficacy in cross-cultural contexts by leveraging advanced techniques for feature extraction and fusion. The principal contributions of this study are as follows:

(1) In this investigation, we introduce TAF-ATRM, a multimodal emotion recognition framework that seamlessly combines text, speech, and image data. Specifically, bidirectional encoder representations from transformers (BERT) is employed to extract deep semantic features from text, Mel-frequency Cepstral Coefficients (MFCC) captures the spectral nuances of speech, and Residual Neural Network (ResNet) is utilized to extract facial expression features from visual modalities. This comprehensive and efficient feature extraction methodology significantly enhances the accuracy and robustness of cross-cultural emotion recognition.

(2) The Transformer model is incorporated into the TAF-ATRM framework, enabling the effective capture of temporal dependencies within multimodal data. Additionally, the multi-head attention mechanism amplifies the framework’s ability to represent features, offering a detailed analysis of complex inter-modal interactions from diverse perspectives, thus markedly improving the multi-level expression and fusion of emotional features.

(3) This article demonstrates that TAF-ATRM surpasses traditional models in the emotion triple classification task, exhibiting superior performance through empirical evaluations on the MOSI and MOSEI datasets. Furthermore, the framework’s applicability in cross-cultural emotion recognition is validated by testing it across diverse cultural contexts, providing substantial support for future endeavors in cross-cultural emotion computation.

The rest of this article is organized as follows: ‘Related Works’ reviews related work on emotion recognition and analysis across different modalities. ‘Methodology’ details the proposed framework. Experimental results and practical evaluations are presented in ‘Experimental Results and Analysis’. ‘Discussion’ offers an in-depth discussion, and the final section concludes the article.

Related works

Sentiment analysis using single modality

Sentiment analysis (Wankhade, Rao & Kulkarni, 2022) involves inferring hidden emotions from data by categorizing them into predefined artificial categories. Traditional sentiment analysis methods, such as text based on sentiment lexicons (Subhashini et al., 2021), rely on calculating sentiment scores by checking whether the text contains specific lexicons from a pre-established dictionary or the polarity of phrases within the text. These lexicons are typically limited to a few thousand words due to the manual creation of sentiment dictionary libraries (Widmann & Wich, 2023). Subhashini et al. (2021) analyzed customer reviews by counting the occurrences of positive and negative words, with the predominant polarity determining the sentiment of the comment. Yadav & Vishwakarma (2020) employed the bag-of-words model to extract textual sentiment features and assessed the effectiveness to sentiment analysis (AlBadani, Shi & Dong, 2022). Yu et al. (2020) constructed a Bi-LSTM model with attention enhancement for speech emotion analysis, achieving high accuracy. Compared to text and speech, facial expressions are more adept at conveying emotions during interpersonal communication, leading to significant research advancements in this domain. Abdat, Maaoui & Pruski (2021) proposed a face key point detection model, which measures changes in facial expressions through 21 distances on the human face. Pei et al. (2025b, 2025a) utilized a CNN-LSTM network to extract spatial features of expressions between frames, applying these features for emotion classification. In recent years, researchers have begun exploring multimodal sentiment analysis methods, integrating multiple information sources—such as text, speech, and video—to obtain a more comprehensive understanding of sentiment.

Sentiment analysis based on multimodal data

Early research in the domain of multimodal sentiment analysis primarily focused on static graphical and textual federated data, with the objective of establishing a mapping relationship between textual-visual features and sentiment semantics to enable sentiment description in social media’s graphical-textual fusion data (Li et al., 2019). With the advent of deep learning, its strong feature representation capabilities have made it the dominant approach in multimodal sentiment analysis. Zadeh et al. (2017) employed a three-fold Cartesian product to fuse multiple unimodal information instead of using simple tensor concatenation. Liu et al. (2018) introduced a multimodal fusion method to enhance the efficiency of multimodal fusion. Some researchers have approached the problem from the perspective of decision fusion. Akhtar et al. (2019) and Ye et al. (2024) decomposed the multimodal sentiment analysis task into several sub-tasks, conducting joint training and performing comprehensive sentiment analysis based on the predictions of each task. Hazarika et al. (2018) used an interactive conversational memory network for multimodal sentiment detection, where global memory is used to capture contextual summaries. Zhang et al. (2019) introduced a quantum-inspired interaction network, combining quantum theory with LSTM networks to learn emotional interactions between discourses. Ding et al. (2023) and Song et al. (2024) employed independent unimodal human annotations to simultaneously learn multimodal and unimodal representations using a multitask learning approach. However, this method requires independent unimodal annotations, and the high difficulty of manual annotation limits its widespread adoption. In response, Yu et al. (2021) proposed generating unimodal annotations automatically through algorithms. While this approach reduces the cost of creating unimodal labels, it is challenged by the instability and unreliability of the automatically generated labels, presenting a new bottleneck in research relying on unimodal emotion labels.

From the above research, it is evident that multimodal data analysis for emotion research across different modalities can effectively address the limitations of sparse data and incomplete information inherent in single-modal emotion analysis. By integrating data at various levels (Lian, Liu & Tao, 2021), the model’s performance is significantly enhanced, enabling more accurate emotion recognition. Additionally, targeted feature improvements within the foundational neural network can further augment model performance. Therefore, in this article, we employ the attention mechanism on top of the original feature fusion to achieve enhanced integration of features enriched with more modal information, facilitating emotion analysis through the fusion of multiple physiological signals.

Methodology

Transformer model and multi head attention

The Multi-Head Attention Mechanism is a pivotal component of the Transformer model, enabling it to execute multiple parallel self-attention computations across different subspaces of the input, thereby allowing the model to attend to various positions or features. This mechanism enhances the model’s representational capacity by performing independent attention operations on multiple “heads” and subsequently concatenating the results. For an input vector X (typically an embedded representation of each word in a sequence), it is projected into three distinct vector spaces, producing matrices of queries (Q), keys (K), and values (V), as depicted in Fig. 1. These scores are divided by a scaling factor dk to balance the range of values, and finally normalized to a probability distribution as follows:

(1) Attention(Q,K,V)=softmax(QKTdk)V

where dk is the key vector dimension. Multiple independent self-attention computations are carried out in parallel by different heads, each utilizing its own set of linear transformation matrices for queries, keys, and values. Specifically, for a given head, the self-attention operation for the I head is computed as follows:

(2) headi=Attention(Qi,Ki,Vi).

Figure 1 The transformer and multi head attention.

The outputs of all the heads are spliced and linearly transformed to get the final multi-head attention output:

(3) MultiHead(Q,K,V)=Concat(head1,head2,…,headh)WO

where WO is the output linear transformation matrix and Concat(⋅) denotes the result of splicing all the heads in dimension. As the attention mechanism continues to evolve, the Transformer model has been introduced and extensively adopted in time series analysis and natural language processing research. The model consists of two primary components: Encoder and Decoder, both structured with multiple layers of self-attention and feed-forward network layers.

The encoder consists of key components such as input embeddings, Multi-Head Self-Attention, a Feed-Forward Network (FFN), residual connections, and layer normalization. The decoder mirrors this structure but adds masking to restrict access to future information during word prediction. Its main components include Masked Multi-Head Attention, Encoder-Decoder Attention, and a Feed-Forward Neural Network, which is made up of two fully connected layers.

(4) FFN(x)=max(0,xW1+b1)W2+b2

where W represents weight matrix, b is the bias vector, and max() is the ReLU activation function. Considering that Transformer does not have a loop mechanism, the order information is introduced by position encoding. The calculation formula for position encoding is.

(5) PE(pos,2i)=sin(pos100002i/dmodel)

(6) PE(pos,2i+1)=cos(pos100002i/dmodel)

where pos is the position, and dmodel is the embedding dimension. Following the decoder, the model uses a linear layer to map the hidden layer representation to a probability distribution over the vocabulary, ultimately generating the target word. Unlike traditional RNN and LSTM models, the Transformer’s parallel processing capability significantly boosts training efficiency. Furthermore, the multi-head attention mechanism enables the model to analyze time series data from multiple perspectives, enhancing its feature extraction capabilities. This allows the Transformer to perform exceptionally well in tasks like time series prediction and classification.

Feature extraction from multimodal data

Text feature

BERT, a pre-trained language model developed by Google, builds upon the Transformer architecture. The key innovation of BERT lies in its bidirectionality, which can understand the semantic information of text through bidirectional context, and has stronger text representation ability compared to traditional recurrent neural network (RNN) or CNN structures. In text classification tasks, BERT can adapt to different fields and tasks through pre training and fine-tuning mechanisms, improving classification accuracy. By taking into account both the preceding and succeeding context of a word within a sentence, BERT demonstrates exceptional effectiveness in language comprehension tasks (González-Carvajal & Garrido-Merchán, 2005). The input to BERT consists of three parts that are token embeddings, segment embeddings, and position embeddings, which will help BERT generate vector representation E(xi) as:

(7) E(xi)=TokenEmbedding(xi)+SegmentEmbedding(xi)+PositionEmbedding(xi)

BERT predicts these masked words by masking a portion of the words in the input. This is a self-supervised learning task. Given input X=[x1,x2,…,xn] the task of BERT is to predict these replaced words.

(8) P(xi∣X\i)=softmax(Whi)

where hi is the output representation of the BERT encoder and W is the output layer weights. BERT also trains the model by predicting the relationship of sentence pairs, with the task of determining whether two sentences given are consecutive or not. Suppose there are two sentences A and B, BERT output vector h[CLS] at [CLS] position is used for classification:

(9) P(IsNext∣A,B)=softmax(Wh[CLS])

where W is the weight of the classifier. In this article, the Transformer model is used to realize its final overall text feature extraction.

Facial feature

For facial image data processing, this article employs the ResNet for facial feature extraction. ResNet’s key innovation lies in its use of residual connections, allowing the network to learn the differences between layers, thus preventing information degradation as it flows through deeper layers. This architecture allows the network to reach considerable depth, significantly enhancing its feature learning capability (Xu, Fu & Zhu, 2023). ResNet (Residual Neural Network) is a deep convolutional neural network that solves the problems of gradient vanishing and exploding in deep neural network training by introducing skip connections. Compared to shallow CNN structures such as visual geometry group (VGG) and AlexNet, ResNet can still maintain good training performance in deeper network structures and learn complex visual features more effectively. In facial expression recognition tasks, expression features are often subtle and require deep networks to extract high-level features. ResNet has strong feature expression ability and can capture local and global information of facial expressions. In addition, ResNet has achieved excellent performance in ImageNet and multiple computer vision tasks, making it more suitable as a backbone network for facial expression recognition compared to traditional CNN structures. Based on this, we chose ResNet to process facial expression data to improve the accuracy and robustness of expression classification. ResNet achieves efficient gradient transfer through residual block structure. ResNet achieves efficient gradient flow through its residual block structure, where each residual block incorporates a skip connection.

(10) y=F(x,{Wi})+x

where x is the input, F(x,{Wi}) is the nonlinear transformation of the convolutional layer (including convolution, bulk normalization, and activation functions), and the residual connection directly adds the input x to the transformed output. By stacking multiple residual blocks, ResNet enables the construction of very deep networks without encountering the issue of vanishing gradients. This architecture allows ResNet to efficiently train networks with hundreds of layers, or even deeper, while preserving high training efficiency. It effectively extracts high-level features in facial expressions, including subtle changes, thereby enhancing the model’s ability to recognize complex expressions.

Audio feature

For the processing of sound signals, this article utilizes both the raw audio signal and the more classical MFCC features for signal extraction. MFCC is a widely used feature extraction method in speech signal processing. Its core concept is to emulate the human ear’s perception of sound by transforming the speech signal into feature coefficients in the frequency domain, allowing it to effectively capture the essential characteristics of speech. MFCC is particularly well-suited for sentiment analysis and audio classification. It is based on the perceptual characteristics of the human ear to different frequencies, mapping speech signals to the Mel frequency scale, thereby more effectively capturing the speech features and timbre information of speech. Compared with traditional time-domain features (such as waveform features) or other frequency-domain features (such as linear predictive cepstral coefficients LPC), MFCC performs better in tasks such as speech recognition and speaker recognition, and can more stably extract key features of speech. In addition, MFCC is computationally simple and has fewer parameters, making it suitable for lightweight computing scenarios. Therefore, we chose MFCC as the speech feature extraction method to improve the representation and classification performance of speech data.

(11) MFCC(n)=∑k=1K⁡log(Sk)cos(πn(k−0.5)K)

where Sk is the energy of the k Mel filter bank, K is the number of filter banks, and n is the index of the MFCC coefficients. MFCC effectively extracts key features from audio by mimicking the perceptual mechanism of the human auditory system, filtering out irrelevant information. This makes it particularly powerful in speech processing, and it is widely applied in tasks such as speech recognition, sentiment analysis, and audio classification.

The establishment for TAF-ATRM model

Following the detailed introduction of each component within the feature extraction module, this article presents the TAF-ATRM network, a multimodal physiological signal emotion recognition framework that integrates text, audio, and facial information. The network enhances the extracted features from text, audio, and facial expressions using the Transformer network, and subsequently fuses these features through the attention mechanism to achieve multi-level feature integration. This approach enables high-precision emotion analysis. The overall structure of the proposed framework is illustrated in Fig. 2.

Figure 2 The framework for TAF-ATRM.

In the TAF-ATRM framework, we adopt a combination of feature stacking and cross modal attention to fully integrate the multimodal information of text, speech, and image data. Firstly, we extract deep features of text, speech, and vision using BERT, MFCC+CNN, and ResNet respectively, and align the features through dimension transformation and temporal alignment to ensure that different modalities are in the same representation space. Subsequently, cross modal attention mechanisms are utilized for information exchange, enabling different modalities to focus on each other and reinforce their key features. Among them, text, speech, and visual features are respectively used as queries, keys, and values, and the correlation between modalities is calculated through multi head attention mechanism to capture their complementary information. After cross modal interaction, the features are concatenated to form a unified high-dimensional representation, which preserves the unique information of each modality and enhances the collaborative effect between modalities. Finally, the fused features are input into a fully connected classifier, which uses Softmax for sentiment classification prediction. Through this fusion strategy, we fully utilize the complementarity of multimodal data and combine it with the self attention mechanism of Transformer to effectively improve the accuracy and performance of sentiment analysis.

Experimental results and analysis

Dataset and experiment setup

In process of dataset selection, this article proposes the use of MOSI and MOSEI, two widely adopted datasets in multimodal sentiment analysis (Zhu, 2023; Wu et al., 2023; Yin et al., 2025). MOSI serves as a benchmark dataset for multimodal sentiment analysis, focusing on sentiment expression in video content. It comprises 93 opinion clips from YouTube, recorded by various speakers, each conveying strong emotional messages. MOSI provides data in three key modalities: textual (the speaker’s verbal content), phonetic (speech features), and visual (facial expressions). MOSEI (Multimodal Sentiment Analysis Dataset) is an expanded version of MOSI, designed for broader and more diverse multimodal sentiment analysis tasks. MOSEI includes video clips from over 1,000 speakers on YouTube, spanning a wide range of topics and sentiment expressions. Like MOSI, it provides data in text, speech, and visual modalities, with each sample labeled for both sentiment and emotion categories (e.g., happiness, anger, sadness). These datasets are valuable for cross-cultural sentiment analysis, as they contain videos from speakers across the globe, encompassing multiple languages, accents, cultural backgrounds, and diverse modes of emotional expression. They provide researchers with the opportunity to explore how emotions are conveyed across cultures, which is crucial for understanding cross-cultural emotion dynamics, especially in the context of multimodal interactions involving language, facial expressions, and speech features. Due to the fact that the MOSI and MOSEI datasets contain speakers from different regions and cultural backgrounds around the world, these videos cover multiple languages, accents, speech patterns, and facial expression differences, making them highly adaptable across cultures. These datasets can effectively reflect the expression of emotions in different cultural backgrounds, enabling the model to learn the commonalities and individualities of cross-cultural emotional features during training, thereby improving its generalization ability in multicultural environments and providing reliable data support for cross-cultural emotion recognition. By utilizing the MOSI and MOSEI datasets, researchers can train models to identify and analyze emotion features in varied cultural contexts, thereby enhancing the accuracy of cross-lingual emotion recognition and improving the comprehension of emotion transfer in cross-cultural communication.

Upon completing the dataset validation, further evaluation of the model’s performance is necessary. In this article, we analyze the results accuracy, precision, recall, and F1-score. We also compare our model against several widely recognized methods, including memory fusion network (MFN) (Zadeh et al., 2018), Recurrent Attended Variation Embedding Network (RAVEN) (Wang et al., 2019), Multimodal Cyclic Translation Network model (MCTN) (Hai et al., 2019), and Multimodal Transformer (MulT) (Delbrouck et al., 2020). Additionally, to investigate the effect of different modal information and feature fusion strategies, we conduct ablation experiments, analyzing variants such as T-ATRM, A-ATRM, and F-ATRM, which utilize only text, audio, or facial modalities, respectively, as well as TAF-TRM, which excludes feature enhancement.

Given that the model processes multimodal data, specific requirements for the experimental environment are necessary. The experimental environment configured for this study is outlined in Table 1.

Table 1 The experiment setup environment.

Environment	Specifications	
CPU	I9-14900K	
GPUs	RTX 4080Ti	
Language	Python 3.9	
Framework	Pytorch	

In light of potential future applications and the role of sentiment analysis in cultural communication, this article focuses on three emotion classifications within the dataset: POSITIVE, NEGATIVE, and NEUTRAL. The aim is to realize sentiment classification within these three categories. The The key hyperparameters for the model is set in Table 2.

Table 2 The key hyperparameters for the model.

Module	Hyperparameter	Value	
Text feature extraction (BERT)	Max sequence length	128	
Hidden dimension	768	
Batch size	32	
Learning rate	2e−5	
Speech feature extraction (MFCC+CNN)	Number of MFCC filters	40	
Frame length	25 ms	
Frame shift	10 ms	
CNN layers	3	
CNN filter size	(3, 3)	
Visual feature extraction (ResNet)	Pretrained model	ResNet-50	
Input image size	224 × 224	
Feature dimension	2,048	
Pretrained weights	ImageNet	
Cross-modal attention (Transformer)	Number of heads (Multi-head attention)	8	
Attention projection dimension	512	
Feedforward network dimension	2,048	
Dropout	0.1	
Feature fusion (Feature stacking)	Concatenation method	Channel-wise concatenation	
Classifier (Fully connected)	Hidden dimension	512	
Classifier (Fully Connected)	Activation function	ReLU	
Classifier (Fully connected)	Learning rate	1e−4	
Classifier (Fully connected)	Optimizer	Adam	
Classifier (Fully connected)	Loss function	Cross-entropy loss	
Classifier (Fully connected)	Batch size	32	

Method comparison and result analysis

After introducing the model, the relevant datasets, and the tasks addressed in this article, we performed an in-depth analysis of the model using two public datasets. We present the variation in loss during the training process, as well as the final recognition accuracy achieved on both datasets. Figure 3 illustrates the training loss process on the MOSI dataset.

Figure 3 Training process and recognition accuracy on the MOSI dataset.

In Fig. 3, it is evident that during the initial stages of training, the loss variation trends across the five models are relatively similar, suggesting comparable learning speeds on the dataset at the outset. However, as the number of iterations increases, the TAF-ATRM framework proposed demonstrates a clear advantage. The loss reduction rate of this framework is significantly faster than that of the other models, indicating that it captures the features in the data more efficiently, thereby optimizing the learning process. Eventually, the loss curve for TAF-ATRM stabilizes, suggesting that the model reaches a more stable state in the later stages of training.

In terms of accuracy for the three-class emotion recognition tasks, the TAF-ATRM framework also exhibits superior performance. The final recognition accuracy stabilizes at 0.863, notably outperforming the other models, demonstrating the higher accuracy and robustness in multimodal sentiment analysis. This framework effectively handles complex sentiment classification tasks. The results for the MOSEI dataset are presented in Fig. 4.

Figure 4 Training process and recognition accuracy on the MOSEI dataset.

In Fig. 4, it is evident that the model’s loss variation trend during training closely correlates with its performance on the MOSI dataset. Given that this dataset contains a larger volume of data and more diverse sentiment expressions, the model’s complexity increases during training, leading to a relatively slower decline in loss. This reflects the challenges faced when handling large-scale, heterogeneous data. Nonetheless, the model proposed continues to exhibit stable performance, ultimately achieving an accuracy of 0.849 on the MOSI dataset. Although this is slightly lower compared to smaller datasets, it still underscores the model’s strong sentiment recognition capabilities on larger datasets. This result further validates the model’s effectiveness in managing complex multimodal sentiment analysis tasks, maintaining a high recognition rate despite the increase in data volume.

To more clearly compare the sentiment recognition performance of different models, we present additional comparison metrics under both datasets. The results of other model performance evaluation metrics are shown in Figs. 5 and 6, as well as in Tables 2–4.

Figure 5 The recognition result on the MOSI dataset.

Figure 6 The recognition result on the MOSEI dataset.

Table 3 The recognition result on MOSI.

Method	Precision	Recall	F1-score	
MFN	0.753	0.763	0.756	
RAVEN	0.759	0.781	0.769	
MCTN	0.772	0.739	0.755	
MulT	0.815	0.795	0.805	
TAF-ATRM	0.859	0.873	0.866	

Table 4 The recognition result on MOSEI.

Method	Precision	Recall	F1-score	
MFN	0.736	0.758	0.747	
RAVEN	0.763	0.771	0.767	
MCTN	0.799	0.783	0.792	
MulT	0.835	0.857	0.847	
TAF-ATRM	0.851	0.849	0.849	

By comparing the three key metrics across the two datasets, it is evident that the TAF-ATRM model performs exceptionally well in the emotion recognition task, demonstrating a strong balance between accuracy and robustness. The experimental results show that the model maintains consistent performance across datasets of varying sizes. On the smaller dataset, the TAF-ATRM model achieves F1-score of 0.866, while on the more complex and large-scale dataset, the F1-score remains high at 0.849. This trend mirrors the accuracy metric, highlighting the model’s stable classification ability and robustness across different dataset scales.

The results further validate the effectiveness of the proposed method, demonstrating its exceptional performance in emotion recognition on smaller datasets while also sustaining high accuracy and consistent results on larger, more complex datasets. This underscores the model’s strong generalization capability and its ability to achieve balanced recognition, even when tackling more challenging multimodal sentiment analysis tasks.

At the same time, we provided the corresponding ROC curves for two models, as shown in Fig. 7.

Figure 7 The ROC cure for the model on both datasets.

According to the results shown in Fig. 7, it can be seen that the overall change of the proposed method in the ROC curve is relatively average. The AUC values of TAF-ATRM in both datasets also change relatively evenly, with overall AUC values of 0.688 and 0.726, respectively. Compared with other models, not only is precision better, but the AUC training process is smoother, which can effectively achieve emotion classification and recognition research.

Ablation experiment and practical test analysis

To more thoroughly analyze the impact of different modal information and the integration of various attention modules on model performance, we conducted ablation experiments. The naming conventions for the models used in these experiments were introduced in “Dataset and experiment setup”. Additionally, to better address the issue of multimodal sentiment analysis in cross-cultural communication, we further extracted data from diverse cultural backgrounds. Specifically, we divided the cultural regions based on IP addresses, and then tested the accuracy of sentiment analysis for individuals from each region using the TAF-ATRM model proposed. This allowed us to compare the framework’s effectiveness across different cultural contexts.

Given the size of the dataset and the necessity for cross-cultural comparison, this section focuses exclusively on the MOSEI dataset. The comparison results of the ablation experiment are provided below, offering insights into how the inclusion or exclusion of specific modal information and attention modules affects model performance across cultures.

In Fig. 8, it is evident that in the single-modal emotion recognition process, even with the addition of the Transformer’s feature enhancement module, the overall emotion recognition performance remains suboptimal. The highest emotion recognition accuracy using only facial expression features is 0.793, falling short of 0.8, while the accuracy achieved with only voice features is 0.697, not exceeding 0.7. In contrast, after fusing the three types of modal information and removing the feature attention module, the model’s overall performance surpasses 0.8 in recognition accuracy. This demonstrates the critical role of multimodal fusion in enhancing emotion recognition.

Figure 8 Ablation experiment result on the MOSEI dataset.

The results for precision, recall, and F1-score in the right subgraph mirror the accuracy outcomes, further validating that different modalities provide essential complementary information during the emotion recognition process. Moreover, the use of attention-enhanced features significantly improves the model’s recognition accuracy, confirming that attention mechanisms contribute to refining feature representations and boosting performance.

After completing the ablation experiments, we conducted a deeper analysis of the TAF-ATRM framework’s ability to recognize emotions across different cultural contexts. To explore the model’s performance in cross-cultural sentiment analysis, we divided the data by continents and compared the model’s detailed performance against classical models and ablation models. As shown in Fig. 9, the accuracy statistics reveal minimal differences in overall model performance across different cultures. However, in terms of sentiment recognition performance, the TAF-ATRM model demonstrates a clear advantage. Whether considering the highest recognition accuracy or the mean accuracy, the TAF-ATRM model consistently outperforms the others.

Figure 9 The recognition result of individuals with different cultural backgrounds.

This indicates that the framework exhibits stronger generalization and robustness when tackling emotion recognition tasks in various cultural contexts. Specifically, TAF-ATRM excels at capturing subtle emotional nuances across cultures while maintaining stable, high performance in diverse data environments. These results suggest that the TAF-ATRM model is not only well-suited to small-scale datasets but also delivers excellent results in large-scale, culturally diverse contexts. This finding offers strong support for future research in cross-cultural sentiment analysis, especially for complex tasks involving global, multicultural scenarios. Furthermore, the TAF-ATRM framework holds significant potential for real-world applications and provides new avenues for enhancing and optimizing cross-cultural sentiment recognition models.

Discussion

In the TAF-ATRM framework proposed, we integrate multi-source data, including text, speech, and images, to capture the essential features of each modality through various feature extraction methods and deep learning models. First, the application of the BERT model in text feature extraction significantly improves the comprehension of complex semantic relationships, particularly by capturing contextual dependencies, thereby preserving and modeling semantic cues in emotional expressions effectively. Second, for speech data, we employ MFCC, which mimics the human auditory system to capture spectral features in speech signals, especially those changes closely tied to emotional expressions. For image data, we utilize the ResNet model for deep visual feature extraction, enabling the capture of subtle emotional information in facial expressions, thus improving the model’s understanding of visual modalities. The key advantage of the TAF-ATRM framework over traditional multimodal sentiment analysis methods lies in its robust feature representation and fusion capabilities. By incorporating the Transformer model and its self-attention mechanism, the framework adeptly handles time-series data, particularly the intricate dependencies across time steps. The multi-attention mechanism further enhances the model’s feature representation by computing multiple sets of self-attentions in parallel, allowing for a more nuanced analysis of the relationships between features from different perspectives. This multi-level feature enhancement enables the extraction of richer emotional cues. Moreover, by stacking multimodal features, the model not only retains the unique characteristics of each modality but also fully leverages the synergistic information between modalities, providing a more comprehensive feature representation for emotion classification. Consequently, compared to traditional methods such as MFN and RAVEN, TAF-ATRM shows significant performance improvements in sentiment classification tasks, particularly in the integration and representation of multimodal features, resulting in higher accuracy and robustness.

By effectively fusing the features of text, speech, and image data, the TAF-ATRM framework enhances the accuracy of sentiment recognition in multi-source data. It excels in managing complex relationships where modalities may complement or contradict each other, leveraging the self-attention mechanism to ensure the overall accuracy of sentiment analysis. This approach fully capitalizes on the complementary nature of multimodal data, addressing the limitations of traditional methods that struggle to capture the complexity of sentiment expression within a single modality. Experimental validation on the MOSI and MOSEI datasets confirms that TAF-ATRM delivers outstanding performance in the sentiment triple classification task, significantly surpassing traditional models. These results demonstrate the framework’s ability to handle the diversity and intricacy of sentiment information in multimodal sentiment analysis tasks. Moreover, the application of the TAF-ATRM framework in cross-cultural sentiment analysis reveals significant potential. Testing across different cultural backgrounds indicates that the framework maintains stable sentiment recognition performance, offering important insights for cross-cultural communication. Emotional expressions vary significantly between cultures, and the TAF-ATRM framework is able to capture these differences and accurately recognize them by fusing multimodal data. This capability provides robust support for emotion understanding in cross-cultural contexts and can facilitate the development of intelligent cross-cultural emotion interaction systems. In addition, the TAF-ATRM framework has broad potential in practical applications, such as cross-cultural customer service systems, intelligent conference translation, international online education, and other scenarios that require precise emotion recognition. For example, in cross-cultural customer service interaction, this framework can help the system accurately identify the emotional state of users, thereby providing more humane services and enhancing user experience. The TAF-ATRM framework holds great promise for future applications in cross-cultural sentiment analysis, especially in tasks involving sentiment understanding and sentiment computation across diverse global cultures. As these tasks become increasingly critical, the framework will play an ever more important role in advancing the field.

Conclusion

In this article, we propose the TAF-ATRM framework, based on the Transformer and Multihead Attention, to address the challenges of feature fusion and temporal dependency in multimodal cross-cultural emotion recognition. By integrating multi-source data such as text, speech, and images, the framework employs BERT to extract textual semantic features, and MFCC and ResNet to extract speech and visual features, respectively, generating high-dimensional multimodal representations. In the emotion recognition process, the framework captures temporal dependencies across different modalities through the Transformer model and further enhances feature representation using the multi-head attention mechanism. Experimental results demonstrate that TAF-ATRM significantly outperforms traditional sentiment analysis methods, such as MFN, RAVEN, and MCTN, on the MOSI and MOSEI datasets. The average recognition accuracy for positive, negative, and neutral sentiment categorization tasks across these datasets is approximately 0.85, reflecting greater accuracy and robustness. In cross-cultural contexts, the model’s mean sentiment recognition efficiency exceeds 0.8, indicating strong utility in diverse cultural environments and verifying the practical value of the framework in multimodal sentiment recognition tasks.

Future research will further expand the application scope of the TAF-ATRM framework and explore its performance in more datasets and complex cross-cultural scenarios. We plan to introduce more types of physiological signals and emotional data, such as neurophysiological signals (EEG, GSR), to enhance the model’s ability to fuse different modal emotions and improve its adaptability in finer-grained emotion recognition tasks. In addition, we will strive for more refined facial expression analysis, focusing not only on basic emotions (such as happiness, anger, sadness), but also expanding to more detailed categories such as micro-expressions, compound emotions (such as bittersweet), and culture-specific expressions, to enhance the accuracy of cross-cultural emotion recognition. By combining real-time feedback mechanisms and adaptive learning, this framework is expected to achieve dynamic tracking and adaptive adjustment of individual emotional states, providing stronger technical support for cross-cultural emotional understanding, intelligent emotional computing systems, and personalized emotional recognition in human-computer interaction. Ultimately, we hope to promote the widespread application of multimodal sentiment analysis techniques in social, educational, mental health, cross-cultural communication, and other fields, enabling them to more naturally understand and respond to emotional expressions in different cultural contexts.

Supplemental Information

Supplemental Information 1 This is the code.

Additional Information and Declarations

Competing Interests

The authors declare that they have no competing interests.

Author Contributions

Xin Bi conceived and designed the experiments, analyzed the data, prepared figures and/or tables, and approved the final draft.

Tian Zhang performed the experiments, performed the computation work, authored or reviewed drafts of the article, and approved the final draft.

Data Availability

The following information was supplied regarding data availability:

The MOSI dataset is available at Zenodo: Wierstorf, H. (2023). Gender annotations for Multimodal Opinion-level Sentiment Intensity dataset (MOSI) [Data set]. Zenodo. https://doi.org/10.5281/zenodo.7554349.

The CMU-MOSEI dataset is available at: https://paperswithcode.com/dataset/cmu-mosei.

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
