# Peer review of "Analysis of the fusion of multimodal sentiment perception and physiological signals in Chinese-English cross-cultural communication: Transformer approach incorporating self-attention enhancement"

_PeerJ Computer Science, doi:10.7717/peerj-cs.2890_

## Round 0.1 · original submission · Major Revisions

Thank you for submitting your manuscript to our esteemed Journal. We appreciate your contribution to the field of multimodal sentiment perception, physiological signals, and cross-cultural communication using Transformer-based models. After a thorough evaluation by our reviewers, we have reached a decision regarding your submission and you will see that a couple of comments/improvements are needed before we proceed further. Please carefully revise the paper in light of these suggestions and submit a detailed response and updated article for further consideration.

AE Comments:
you mention BERT for text, MFCC for speech, and ResNet for facial expressions, it would be beneficial to justify why these specific models were chosen over alternatives (e.g., OpenSmile for speech, EfficientNet for facial recognition)
Elaborate how text, speech, and facial image data are preprocessed before being fed into models would enhance reproducibility.

The study claims that TAF-ATRM outperforms traditional methods, but a direct comparison with other state-of-the-art models (e.g., LSTM-based models, CNN-based fusion networks) would strengthen the results

Improve the technical language of the paper.

**Language Note:** The Academic Editor has identified that the English language must be improved. PeerJ can provide language editing services - please contact us at [email protected] for pricing (be sure to provide your manuscript number and title). Alternatively, you should make your own arrangements to improve the language quality and provide details in your response letter. – PeerJ Staff

·

Basic reporting

The manuscript introduces a multimodal sentiment perception model (TAF-ATRM), incorporating BERT, MFCC, ResNet, and Transformer-based self-attention for sentiment recognition in cross-cultural contexts. The study is well-structured and presents promising results, but several aspects need further elaboration, especially regarding fusion strategies, interpretability, and cross-cultural validation.
- The model extracts features from text, speech, and facial expressions, but the fusion strategy between modalities remains unclear.
- The paper assumes that multimodal sentiment perception varies across cultures, but does not provide linguistic or psychological theories supporting this claim.
- Cite works from Hofstede’s cultural dimensions theory or Edward Hall’s high-context/low-context communication frameworks to substantiate why multimodal emotion expression differs across cultures.
- Add a table summarizing training parameters and provide details on how hyperparameters were optimized (e.g., grid search, Bayesian optimization).
- Conduct an additional ablation study that removes:
Transformer’s self-attention (replaced with Bi-LSTM).
Multi-head attention mechanism (using simple concatenation).
BERT-based text features (using word embeddings like GloVe).
- The manuscript primarily reports accuracy and F1-score, but real-world sentiment systems require robustness measures like ROC-AUC or calibration metrics. Provide additional performance metrics, including:
Confusion matrices to highlight classification errors.
ROC-AUC curves to evaluate model discrimination ability.
- Figures 3, 4, and 5 are visually cluttered with small fonts and overlapping curves. Improve figure clarity by:
Using distinct color schemes for different models.
Enlarging axis labels and legends for readability.

Experimental design

.

Validity of the findings

.

Additional comments

Below are some specific sections comments

Abstract Section: Abstract should be vigorous while having the short introduction, contribution, methodology, experimental setup, study criteria, comparison of results and concluding remarks.
Inadequate Introduction: Introduction should include proper background of the research area and research topic as well as the works from state of the arts.
Graphics and Visual Aids: Some of the figures (1, 2 and 7) in the manuscript could be enhanced for clarity and better understanding. You should redraw the figures extracted from literature or mention the reference of literature with each figure.
Explanation of Figures and Tables: Additional explanations or captions for all figures would enhance their relevance and clarity in the context of the text.
Formatting and Structure: There are some inconsistencies in the formatting (font, line spacing, headings) and structure of the manuscript that could be streamlined for a more professional presentation.
Technical Terminology: The use of technical jargon is at times inconsistent. Ensuring consistent use of technical terms would improve the manuscript's readability. E.g. “TAF-ATRM”, “MFCC”, “MOSI”, “Bi-LSTM”.
References and Citations: The manuscript should include more recent references, particularly from the last two years, to ensure the literature review is up-to-date. As we as there should be proper format of references.

Reviewer 2 ·

Basic reporting

The TAF-ATRM model is technically sound and leverages Transformer-based feature fusion effectively. However, improvements are needed in explainability, scalability, and computational efficiency.
- Add a discussion on the linguistic and cultural biases in the datasets and justify how the model remains robust across cultures.
- The study does not mention practical deployment scenarios for TAF-ATRM. Outline potential applications, such as:
- International customer service chatbots.
- Real-time emotion monitoring in cross-cultural business negotiations.

Experimental design

- Provide a clear equation or schematic detailing how multi-head attention combines information across modalities. Also, explain whether fusion occurs at the feature level or decision level.
- Experiment with mBERT (Multilingual BERT) or XLM-R and discuss if language translation introduces sentiment misinterpretations.

Validity of the findings

- Either include comparisons with more advanced multimodal fusion models or justify why the chosen baselines are representative.
- Apply SHAP (SHapley Additive Explanations) or attention heatmaps to visualize which features (text/speech/visual) contribute most to the sentiment prediction.
- Sentiment perception changes with context (e.g., sarcasm, politeness markers). Discuss how the model handles sentiment modifiers, such as negations ("not happy") or intensifiers ("very happy").

---

## Round 0.2 · accepted · Accept

Based on the input from the experts on your revision, I'm pleased to inform you that your manuscript is judged scientifically sound to be recommended for publication. I do agree with the reviewers opinion. Congratulations and thank you for your fine contribution

·

Basic reporting

All comments are incorporated...

Experimental design

.....

Validity of the findings

.......

Reviewer 2 ·

Basic reporting

no comment

Experimental design

no comment

Validity of the findings

no comment

Additional comments

no comment